Foundation species loss affects vegetation structure more than ecosystem function in a northeastern USA forest

Orwig David A. 1
Barker Plotkin Audrey A. 1
Davidson Eric A. 2
Lux Heidi 1
Savage Kathleen E. 2
Ellison Aaron M. aellison@fas.harvard.edu 1
1 Harvard University, Harvard Forest , Petersham, MA , USA
2 Woods Hole Research Center , Falmouth, MA , USA
Huston Michael
Electronic publication date: 2013 Feb 19
Publication date: 2013
Volume: 1
Electronic Location ID: e41
Received 2012 Nov 13; Accepted 2013 Jan 28
Copyright: © 2013 Orwig et al.
Copyright year: 2013
Copyright holder: Orwig et al.
License: This is an open access article distributed under the terms of the Creative Commons Attribution License, which permits unrestricted use, distribution, and reproduction in any medium, provided the original author and source are credited.
License URL: https://creativecommons.org/licenses/by/3.0/

Keywords: Adelges tsugae, Eastern hemlock, Hemlock woolly adelgid, Primary productivity, Logging, Nutrient cycling, Soil respiration, Tsuga canadensis, Forest dynamics

Funding: US NSF 06-20443 02-36897 04-52254 US DOE DE-FC02-03-ER63613 DE-FC-02-06-ER64157 This research was supported by the US National Science Foundation through its programs in Long Term Ecological Research (DEB 06-20443), Ecosystems Research (DEB 02-36897), and Research Experiences for Undergraduates (DBI 04-52254), and by the US Department of Energy’s Office of Science through its Northeast Regional Center of the National Institute for Global Environmental Change under cooperative agreement no. DE-FC02-03ER63613. The funders had no role in study design, data collection and analysis, decision to publish, or preparation of the manuscript.

==============================
Loss of foundation tree species rapidly alters ecological processes in forested ecosystems. Tsuga canadensis, an hypothesized foundation species of eastern North American forests, is declining throughout much of its range due to infestation by the nonnative insect Adelges tsugae and by removal through pre-emptive salvage logging. In replicate 0.81-ha plots, T. canadensis was cut and removed, or killed in place by girdling to simulate adelgid damage. Control plots included undisturbed hemlock and mid-successional hardwood stands that represent expected forest composition in 50–100 years. Vegetation richness, understory vegetation cover, soil carbon flux, and nitrogen cycling were measured for two years prior to, and five years following, application of experimental treatments. Litterfall and coarse woody debris (CWD), including snags, stumps, and fallen logs and branches, have been measured since treatments were applied. Overstory basal area was reduced 60%–70% in girdled and logged plots. Mean cover and richness did not change in hardwood or hemlock control plots but increased rapidly in girdled and logged plots. Following logging, litterfall immediately decreased then slowly increased, whereas in girdled plots, there was a short pulse of hemlock litterfall as trees died. CWD volume remained relatively constant throughout but was 3–4× higher in logged plots. Logging and girdling resulted in small, short-term changes in ecosystem dynamics due to rapid regrowth of vegetation but in general, interannual variability exceeded differences among treatments. Soil carbon flux in girdled plots showed the strongest response: 35% lower than controls after three years and slowly increasing thereafter. Ammonium availability increased immediately after logging and two years after girdling, due to increased light and soil temperatures and nutrient pulses from leaf-fall and reduced uptake following tree death. The results from this study illuminate ecological processes underlying patterns observed consistently in region-wide studies of adelgid-infested hemlock stands. Mechanisms of T. canadensis loss determine rates, magnitudes, and trajectories of ecological changes in hemlock forests. Logging causes abrupt, large changes in vegetation structure whereas girdling (and by inference, A. tsugae) causes sustained, smaller changes. Ecosystem processes depend more on vegetation cover per se than on species composition. We conclude that the loss of this late-successional foundation species will have long-lasting impacts on forest structure but subtle impacts on ecosystem function.

Introduction

Changes in the distribution and abundance of canopy trees have system-wide impacts on ecological processes in forests (Lovett et al., 2006; Wardle et al., 2011; Hicke et al., 2012). Changes in species composition and associated ecological impacts also lead to changes in the values – including economic, utilitarian, and aesthetic – that we place on forest ecosystems (e.g., Aukema et al., 2011; Cardinale et al., 2012). The vast majority of studies of the impacts of species loss on ecological processes in forests and other ecosystems have examined how changes in the absolute number (or percent) of species lost affects a wide range of ecosystem services (recently reviewed by Wardle et al., 2011; Cardinale et al., 2012; Hooper et al., 2012; Naeem, Duffy & Zavaleta, 2012). However, species are not lost from ecosystems at random (e.g., Bunker et al., 2005) and it remains an open question whether particular species with particular characteristics will disproportionately change how ecosystems function (Bunker et al., 2005; Suding et al., 2008; B Baiser & AM Ellison, unpublished data).

Foundation species (sensu Ellison et al., 2005a) define and structure many terrestrial, aquatic, and marine ecosystems, yet because foundation species often are abundant and widespread, their role in structuring ecosystems is often underappreciated or taken for granted, and they are rarely of explicit conservation interest (Gaston, 2010). Ellison et al. (2005a) suggested that the loss of foundation species can cause strong, widespread, and long-lasting changes to forest ecosystems because forest-wide biological diversity and ecosystem processes such as primary productivity and fluxes of energy and nutrients are hypothesized to depend more on foundation species than on any other species in the system. Examples where loss of dominant, and possibly foundational, tree species have had large impacts on forest ecology include: regional loss of associated fauna as white pines (Pinus subgenus strobus) in western North America succumb to white pine blister rust (Cronartium ribicola A. Dietr.); changes in canopy structure as a result of fire suppression, irruptions of mountain pine beetle (Dendroctonus ponderosae Hopkins), and climatic change (Kendall & Keane, 2001; Tomback & Achuff, 2010); shifts in understory composition, recruitment, and regeneration dynamics following loss of American beech (Fagus grandifolia Ehrh.), American chestnut (Castanea dentata (Marsh.) Borkh.) or American elm (Ulmus americana L.) due to beech-bark disease (Nectria coccinea (Pers. ex. Fr.) Fries var. faginata Lohman, Watson and Ayers), chestnut blight (Cryphonectria parasitica (Murrill) Barr.), and Dutch elm disease (Ceratocystis ulmi (Buism.) C. Moreau), respectively (McBride, 1973; Houston, 1975; Barnes, 1976; Huenneke, 1983; Twery & Patterson, 1984; Myers, Walck & Blum, 2004; Lovett et al., 2006); changes in faunal (Wills, 1993) and macrofungal diversity (Anderson et al., 2010), and functional diversity of soil bacteria involved in carbon and nitrogen cycling (Cai et al., 2010) following loss of Eucalyptus to Phytophthora outbreaks in Australia; bottom-up control by Populus spp. of associated herbivorous arthropod populations, which in turn mediates how insectivorous birds influence future tree growth in the southwestern United States (Bridgeland et al., 2010); and the dependence of benthic biological diversity, productivity, and nutrient cycling on a handful of species in mangrove forests (e.g., Nagelkerken et al., 2008; Barbier et al., 2011).

Tsuga canadensis (L.) Carr. (eastern hemlock), an hypothesized foundation tree species (Ellison et al., 2005a), covers ≈10 000  km2 and comprises ≈ 2 × 108 m3 of harvestable and merchantable volume from the southern Appalachian Mountains north into southern Canada and west across the upper Midwestern states in North America (Fig. 1; Smith et al., 2009). Like other putative foundation tree species, T. canadensis can account locally for >50% of the total basal area, and its ecological traits create unique terrestrial and aquatic habitats. For example, the deep shade cast by its dense evergreen foliage limits establishment of most understory species (Rogers, 1980; D’Amato, Orwig & Foster, 2009). Its refractory leaf litter and the cool temperatures at the soil surface beneath dark hemlock canopies result in low rates of decomposition and nutrient cycling, rapid accumulation of organic matter (Aber & Melillo, 1991; Jenkins, Aber & Canham, 1999), and nutrient-poor soils. The combination of nearly year-round low photosynthetic and evapotranspiration rates of T. canadensis (Hadley et al., 2008)  stabilizes stream base-flows and decreases daily variation in stream temperatures (Ford & Vose, 2007; Nuckolls et al., 2009). The microhabitat created by eastern hemlock supports unique assemblages of birds, arthropods, salamanders, and fish (Snyder et al., 2002; Tingley et al., 2002; Ellison et al., 2005b; Dilling et al., 2007; Mathewson, 2009; Rohr, Mahan & Kim, 2009; Mallis & Rieske, 2011; Sackett et al., 2011).

Despite its widespread distribution and high abundance, both locally and regionally, T. canadensis is rapidly disappearing across an increasing extent of its range. The hemlock woolly adelgid (Adelges tsugae Annand), an invasive insect from Japan that in North America feeds exclusively on eastern hemlock and its southeastern (USA) endemic congener, Carolina hemlock (T. caroliniana Engelmann), is moving rapidly both southward and northward (Fitzpatrick et al., 2012), killing >90% of hemlocks it encounters (Orwig, Foster & Mausel, 2002; Eschtruth et al., 2006; Knoepp et al., 2011). Hemlock has little resistance to the adelgid (Ingwell & Preisser, 2011) and as yet has shown no recovery from chronic infestations (McClure, 1995; Orwig et al., 2012). In the absence of successful biological control programs (Onken & Reardon, 2011) and economically or logistically feasible chemical control options (Ward et al., 2004; Cowles, 2009), pre-emptive cutting or salvage logging of hemlock has been a common management response to declining and dead hemlock stands affected by the adelgid (Kizlinski et al., 2002; Orwig, Foster & Mausel, 2002; Ward et al., 2004; Foster & Orwig, 2006).

The combination of adelgid-induced morbidity and mortality, and pre-emptive salvage logging of T. canadensis is radically changing the structure of eastern USA forests. Region-wide, forest productivity and carbon sequestration are expected to decline by as much as 8%–12%, but establishment of mid-successional hardwoods (e.g., Betula and Acer species) is forecast to result in forest carbon uptake recovering to, or even exceeding pre-adelgid conditions only after 50 years or more (Albani et al., 2010; Knoepp et al., 2011; Fitzpatrick et al., 2012). These model forecasts of the impact of the adelgid have been made at coarse-grained scales (2.5° grid), but local impacts may fall short of or dramatically exceed regional averages (PC Lemos & AC Finzi, unpublished data). Fifteen years of observational studies of marked plots have illustrated high variance in forest dynamics (e.g., Orwig, Foster & Mausel, 2002; Orwig et al., 2008; Orwig et al., 2012), portions of which may be attributable to differences in climate, short- versus long-term impacts of logging, and/or fine-scale effects of the adelgid itself (Stadler et al., 2005; Stadler, Müller & Orwig, 2006).

Only experimental studies can distinguish reliably among differences due to in situ forest disintegration or logging, and so in 2003 we established a multi-hectare, long-term manipulative study – the Harvard Forest Hemlock Removal Experiment (HF-HeRE; Ellison et al., 2010) – to study the various forest responses to the loss of hemlock. This ongoing experiment compares and contrasts the rates, magnitudes, and trajectories of changes in hemlock-dominated stands to two mechanisms of foundation species loss: (1) death in place of eastern hemlock by girdling, which mimics tree disintegration that follows infestation by the hemlock woolly adelgid (Yorks, Leopold & Raynal, 2003); or (2) loss and removal of hemlock following commercial logging (Brooks, 2001). Patterns, processes, and dynamics studied include: forest vegetation structure, standing and downed dead wood, and three measures of ecosystem function: litterfall (a substantial component of net primary productivity; e.g., Zheng, Prince & Hame, 2004), soil carbon flux, and soil nitrogen dynamics.

In this paper, we report two years of pre-treatment data and the first five years of changes in vegetation structure and ecosystem functions following our experimental manipulations but prior to the infestation of our experimental plots by the adelgid. In particular, we examine and test three predictions that, relative to both hemlock and hardwood controls: (1) Vegetation structure – species richness and cover of understory herbs, and density and cover of tree seedlings and saplings – increases slowly following girdling but more rapidly following hemlock removal and soil scarification from logging;

(2) Volume of standing dead wood and snags is highest in girdled plots, but downed coarse woody debris is higher in logged plots;

(3) Core ecosystem functions – litterfall and soil carbon fluxes decline while rates of soil nitrogen (as nitrate and ammonium) mineralization and soil nitrogen availability increase then decline slowly in girdled plots but rapidly in logged plots.

Other papers have described changes in the microenvironment (Lustenhouwer, Nicoll & Ellison, 2012), species composition of the seed bank and understory vegetation (Sullivan & Ellison, 2006; Farnsworth, Barker Plotkin & Ellison, 2012), diversity of ground-dwelling arthropods (Sackett et al., 2011), and nitrogen leaching (Templer & McCann, 2010) in the first decade following the canopy manipulations in HF-HeRE. In total, our results lead us to hypothesize that vegetation structure and ecosystem functions in the girdled and logged plots will converge through time, and, at least on decadal scales, come to resemble the attributes of the hardwood control plots.

We note that we purposely sited HF-HeRE north of the northern limit (in 2003) of the hemlock woolly adelgid so that we could first identify different effects on forest structure and function caused by two different kinds of physical loss of T. canadensis. This experiment complements a suite of studies in which we have examined landscape-level spread of the adelgid (Orwig, Foster & Mausel, 2002; Fitzpatrick et al., 2012; Orwig et al., 2012), compositional and structural changes in forest vegetation (Orwig & Foster, 1998), and ecosystem functions in forests infested by the adelgid (Cobb, Orwig & Currie, 2006; Orwig et al., 2008) or that have been salvage logged (Kizlinski et al., 2002, DA Orwig et al., unpublished data). Subsequent data collected after the adelgid colonizes HF-HeRE (which occurred in 2010), will be used to further distinguish effects on eastern North American forests of physical disintegration of T. canadensis from additive, interactive, and/or nonlinear effects of the insect itself (e.g., Stadler et al., 2005; Stadler, Müller & Orwig, 2006). The unique experimental design – with measurements made pre-treatment; post-treatment but pre-adelgid; and post-treatment, post-adelgid – distinguishes HF-HeRE from other studies, both observational and experimental, that have examined the effects of foundation species loss but that cannot separate effects of physical loss alone from those of the agent of loss itself.

Materials and Methods

Site Description

HF-HeRE is located within the 121-ha Simes Tract (42.47°–42.48°N, 72.22°–72.21°W; elevation 215–300 m a.s.l.) at the Harvard Forest Long Term Ecological Research Site in Petersham, Massachusetts, USA (Ellison et al., 2010, Fig. 1). As in most New England forests, the Simes Tract as was cleared for agriculture in the early and mid-1800s. Many of the trees that had regenerated following agricultural abandonment in the mid- to late-1800s were blown down in the 1938 Great Hurricane, and analysis of tree-cores from the tract show that the trees in our experimental plots average 55–80 years old (Bettmann-Kerson, 2007; AM Ellison, DA Orwig & AA Barker Plotkin, unpublished data), The soils are predominantly coarse-loamy, mixed, active, mesic Typic Dystrudepts in the Charlton Series that are derived from glacial till (USDA). Across the eight HF-HeRE study plots, the soil pH ranges from 3.0–3.4 in the organic layer and from 3.5–4.0 in the mineral layer, and the soil C:N ratios range from 26–33. Much of the central portion of the tract is poorly drained or swampy; elevated areas are better drained. Tsuga canadensis and Acer rubrum L. (red maple) dominate the poorly drained soils, whereas T. canadensis, along with Quercus rubra L. and Q. alba L. (red and white oaks), and Pinus strobus L. (white pine) predominate on hills and slopes. Betula lenta L. (black birch), Acer saccharum Marsh. (sugar maple), and other hardwoods grow at low frequency and density throughout the tract (Ellison et al., 2010).

Experimental design and treatments

The complete design of HF-HeRE is described by Ellison et al. (2010); only salient details are repeated here. The eight 90 × 90 m (0.81 ha) plots comprising this experiment are grouped in two blocks (Fig. 1), each consisting of three plots initially dominated by T. canadensis and one plot of mixed hardwoods (Table 1). The “valley” block (plots 1–3 and 8) is in undulating terrain bordered on its northern edge by a Sphagnum-dominated wetland (permission to work in this wetland and in the adjacent bordering vegetation [“buffer zone”] was provided by the Petersham, Massachusetts, Conservation Commission). The “ridge” block (plots 4–7) is on a forested ridge. Plots were identified in 2003 and sampled for two growing seasons (spring/summer in each of 2003 and 2004) prior to applying canopy manipulation treatments – girdling, or harvesting of standing T. canadensis along with cutting of merchantable hardwoods and P. strobus – to one plot in each block.

Figure 1 Location of the Harvard Forest Hemlock Removal Experiment.

Location of the Harvard Forest Hemlock Removal Experiment in Massachusetts, USA. The regional map shows the basal area of eastern hemlock at a 1 km2 resolution. The inset shows the location of the experimental blocks and treatments. Plots 1, 2, 3, and 8 make up the valley block; plots 4–7 make up the ridge block. Each canopy manipulation treatment – hemlock control (He), girdled (G), logged (L), and hardwood control (Hw) – was applied to a 90 × 90 m plot within each block.

Table 1 Changes in total average basal area (m2 ha−1) and density (ha−1) in the treatment plots of the Harvard Forest Hemlock Removal Experiment.

		Basal area	Stem density	
Canopy manipulation	Year	Valley plots	Ridge plots	Valley plots	Ridge plots	
Hemlock control	2004	45.6	52.1	940	678	
2009	47.3	54.0	842	637	
Girdled	2004	50.3	53.0	1354	1011	
2009	15.9	17.6	395	331	
Logged	2004	47.9	49.5	1212	1089	
2009	15.4	13.8	469	373	
Hardwood control	2004	29.7	35.6	1122	885	
2009	31.0	37.7	990	807	

In the girdled treatment plots, the bark and cambium of all individual T. canadensis trees, saplings, and seedlings were cut through using chain saws or hand knives over a 2-day period in May 2005. Girdled trees died over a 2-year period but were left standing in place to simulate the physical decline and mortality of hemlock resulting from infestation by the hemlock woolly adelgid (Ellison et al., 2010). No other species were girdled and there was no site disturbance other than walking between trees.

In the logged treatment plots, all T. canadensis trees >20 cm diameter at breast height (DBH, measured 1.3 m above ground) and 50% of the commercially valuable Q. rubra and P. strobus were felled using a chainsaw and removed using a skidder between February and April 2005, when the ground was frozen. Because this logging operation mimicked the effects of an intensive commercial hemlock salvage operation, trees of small size, poor quality, or little economic value, such as A. rubrum and B. lenta were also removed to facilitate log removal or to improve future stand quality, but some good-quality Q. rubra and P. strobus were retained. We recognize that the removal of tree species other than T. canadensis can have some impacts on changes forest dynamics in the otherwise hemlock-dominated stands. However, in the logged plots, T. canadensis accounted initially for >50% of the basal area, but made up >80% of both the number of felled trees and their basal area (Ellison et al., 2010). Thus, the effects of hemlock loss were likely to dominate observed responses of the forest to this canopy manipulation.

Two control plots in each block were not manipulated. In each block, one of each of these control plots was dominated by hemlock, the other by mid-successional hardwoods of the same general age of the remaining forest (55–80 years). The latter control plots represents the most likely future forest conditions after hemlock has disappeared from the landscape (Orwig & Foster, 1998; Ellison et al., 2010).

Measurements

Vegetation structure

We measured species richness and cover of understory herbs, and density and cover of tree seedlings and saplings to determine how these attributes of vegetation structure varied among the two canopy manipulation treatments and the two different controls (Prediction 1). In 2003 (prior to canopy manipulations), we established two transects running through the central 30 m × 30 m of each plot to quantify understory richness, cover, and density. Five 1- m2 subplots were spaced evenly along each transect and have been sampled annually since 2003. In each subplot, tree seedlings (<1.3 m tall) were counted and percent cover of tree seedlings, herbs, shrubs, ferns, and grasses was estimated to the nearest one percent. Grasses and sedges were identified only to genus as most lacked flowers or fruits necessary for accurate species-level identification. A species list has been compiled annually for the central 30 × 30-m core area of each plot. Nomenclature follows Haines (2011). The number of sapling-sized trees (>1.3 m tall but <5 cm DBH) was tallied by species in the 30 × 30-m core area of each plot in 2005, 2007 and 2009.

In 2003–2004, all trees ≥ 5 cm DBH in each plot were tagged with aluminum tags, identified, measured (DBH) and mapped (x, y, z coordinates relative to a plot corner) using a compass, auto-level, and stadia rods. Initial basal area was higher in the hemlock plots (45.6– 53 m2 ha−1) than in the hardwood control plots (29.7– 35.6 m2 ha−1) and basal area was slightly higher in the ridge block than in the valley block (species composition data reported in Table 1 of Ellison et al. (2010)). Initial stem density ranged from 678 stems ha−1 in the ridge hemlock control plot to 1354 stems ha−1 in the valley girdled plot. Tsuga canadensis comprised 50%–69% of initial basal area and 55%–70% of initial stem density in the six plots initially dominated by this species (hemlock control, logged, and girdled plots). In the hardwood control plots, T. canadensis comprised <10% of the initial basal area and 10%–11% of the initial stem density. Other species that comprised >10% initial basal area in any plot included A. rubrum, A. saccharum (hardwood valley plot only), B. lenta, Q. rubra, and P. strobus. Decline and death of trees in the girdled plots was tracked following treatments; most T. canadensis had died within 24 months (Ellison et al., 2010). In 2009, each tree was assessed for survival and diameter growth (for living stems).

Standing and downed dead wood

Prediction 2 addresses changes in volume of coarse woody debris (CWD) – standing dead trees (snags), stumps, and volume of fallen boles and branches >7.5 cm in diameter – as a function of canopy manipulation treatment. These variables have been surveyed biennially since 2005 (post-treatment only). In summer 2005, just after the girdling and logging were completed, eight transects were established in each plot beginning from each cardinal and ordinal plot corner/edge and extending 35–50 m towards the plot center. To measure standing dead wood, snags and stumps were sampled along a 4-m wide strip plot that straddles the line transect. Species (or species group) were recorded for each individual stump or snag; its lower diameter was measured, and its top diameter and height either were measured directly or estimated if the snag height exceeded the length of a stadia rod. From these measurements, snag volume was calculated as the frustum of a cone (Harmon & Sexton, 1996). Volume of downed wood was estimated using the line-intercept method (Van Wagner, 1968). The diameter, decay class (Rice et al., 2004), and species (or species group) of each piece that intersected the line was recorded.

Ecosystem function

Prediction 3 is that primary productivity, soil carbon flux, and soil nutrient cycling and availability should decline slowly in the girdled plots and rapidly in the logged plots. We used litterfall as an index of annual aboveground productivity (e.g., Zheng, Prince & Hame, 2004). Five litterfall baskets (collection area 0.11 m2) were placed at random coordinates throughout each 90 × 90-m plot. Baskets were placed in the field at the beginning of September 2005 (after canopy manipulations had been applied). Samples were collected quarterly: in early April, mid-June, mid-September, and early December every year. Leaf litter was sorted to major species groups (Tsuga, Pinus, Quercus, Betula, Acer, other deciduous trees), whereas twigs, bark, and reproductive parts were pooled into one category. After sorting, samples were oven-dried at 70 °C for 48 h prior to weighing. Annual litterfall is reported as the total of the June, September and December collections, plus the subsequent year’s April collection.

Measurements of soil carbon (C) flux (“soil respiration”) were made using a vented, flow-through, non-steady-state system (Livingston & Hutchinson, 1995) at six randomly chosen locations in the 30 × 30-m core area of each of the six hemlock (control, girdled, logged) plots (2003–2009) and at two randomly chosen locations in the two hardwood control plots (2006–2007). At each location, soil respiration collars, each 25 cm in diameter (0.05 m2 surface area) and made from thin-walled polyvinylchloride (PVC) tubing cut into 10 cm lengths, were inserted ≈5 cm into the ground. Soil respiration was measured manually every 2 weeks during the growing season between 09:00 and 15:00 h using a Li-Cor 6252 portable Infrared Gas Analyzer (IRGA) (Li-Cor Inc., Lincoln, Nebraska, USA) mounted on a backpack frame. The IRGA was connected to a vented white acrylonitrile-butadiene-styrene (ABS) chamber top (10 cm in height) that was then placed over the soil respiration collar. A pump circulated the air at 0.5 L min−1 from the chamber top through the IRGA and back to the chamber top. The chamber top was left on the collar for 5 min, and the change in CO2 concentration within the chamber was recorded using a Hewlett-Packard HP 200LX palmtop computer (Hewlett-Packard, Palo Alto, California, USA). The calibration of the IRGA was checked each day that measurements were made using both zero mL L−1 CO2 and 594 mL L−1CO2 certified standards. A linear regression of concentration of CO2 versus time was used to determine the soil respiration rate, which was then corrected for local atmospheric pressure and chamber air temperature. The response variable used in subsequent analyses of treatment effects was soil respiration for the entire sampling period (Day of Year [DOY] 191-273) each year; this value was estimated by linearly interpolating soil respiration measurements between sampling days for each collar and then summing (integrating) all values over the 82-day sampling period.

Total soil respiration is the sum of two belowground components: heterotrophic (microbial and microfaunal respiration) and autotrophic (root respiration). Measurements of soil respiration in the control plots represent the sum of these belowground processes. Thus, to a first approximation, differences in soil respiration between control and either logged or girdled treatments reflect the contribution of eastern hemlock to autotrophic respiration. Decreased soil respiration due to treatment effects were calculated by taking the pre-treatment soil respiration over the sampling season and subtracting from it the post-treatment seasonal soil respiration. These decreases represent a conservative estimate of autotrophic soil respiration in treated plots. Potential limitations to this method include the loss of root biomass, which could reduce heterotrophic respiration of soil organic matter via lack of priming, and that the newly severed roots may temporarily increase carbon available for heterotrophic respiration.

Because the majority of live tree roots in each plot were killed following logging or girdling of hemlock, and because the percent cover of other vegetation in these treatments at the beginning of this study was very low (<2%), seasonal sums of soil respiration in these canopy manipulation treatments can be used as estimates of heterotrophic soil respiration (Hanson et al., 2000; Levy-Varon, Schuster & Griffin, 2012).

Nitrogen (N) mineralization measurements were begun in August 2003, two growing seasons prior to canopy manipulations, using a modified core method (Hart et al., 1994; Roberston et al., 1999). In the central 30 × 30-m area of each canopy manipulation plot, closed-topped cores were installed within four, 5 × 5-m, randomly located subplots each year at 7-week intervals during the growing season (May–October), and for a 23-week overwinter (October/November–April) incubation. At the beginning of each sampling period, soil was extracted with sharpened PVC cores (25-cm long) and immediately separated into mineral and organic layers. A second core was incubated in the field for 42–50 days and then removed and separated by horizon. The bottom 2 cm of each core was removed to prevent root invasion from below in incubated cores, and to standardize sample volume among the cores.

Soil samples were returned to the laboratory on ice and processed the next day. Organic and mineral soils were passed through a 5.0-mm mesh screen, weighed for total mass, and subsampled for gravimetric moisture and inorganic N. To determine soil NH4–N and NO3–N concentration, ∼ 10 g of organic and mineral soil were placed into 100 ml of 1 M KCl for 48 h (Aber et al., 1993). Soil extracts were filtered through a coarse pore filter (0.45– 0.6 µm) and inorganic N concentrations were determined colorimetrically with a Lachat 8500 flow-injection autoanalyzer (Lachat Instruments, Inc., Milwaukee, Wisconsin, USA), using the salicylate (Lachat Instruments Inc., 1990a) and cadmium reduction methods (Lachat Instruments Inc., 1990b) for NH4–N and NO3–N, respectively. Net N mineralization was calculated as the difference in concentration of inorganic N (NH4–N + NO3–N) in incubated cores minus that in initial samples.

An additional assessment of forest floor N availability and mobility was determined at each soil subplot using mixed-bed cation + anion resin bags (Binkley & Matson, 1983). Approximately 10 g of resin was placed in nylon mesh bags and pretreated with 2 M KCl before deployment for 6-month intervals (growing season and overwinter). Resins were deployed at the forest floor – mineral soil interface within 5 cm of where the N mineralization incubations were located. Resins were returned to the laboratory on ice, dried at 105 °C for 24 h, and extracted in 2 M KCl. Inorganic N was determined by the methods described above for soil N extracts.

Statistical analysis and data availability

The experimental design is a one-way blocked analysis of covariance (ANCOVA) (Ellison et al., 2010; Gotelli & Ellison, 2012), and analyses reported here were executed using the lme function in the nlme package in R version 2.9.2 (R Development Core Team, 2009; Pinheiro et al., 2012). In this design, the four canopy manipulations (hemlock control, hemlock girdled, logged, hardwood control) were treated as “fixed” factors, the two blocks were treated as “random” factors, and time entered the model as a covariate. Measures of vegetation structure and ecosystem function parameters were log-transformed as needed to normalize data and equalize variances; data are plotted back-transformed (Gotelli & Ellison, 2012). Comparisons among treatments were done using a priori contrasts. Although two blocks is the minimum required to allow for estimates of variance within treatments, this small number of blocks provided relatively low statistical power to detect true differences among treatments (i.e., the probability of a Type II error – falsely accepting the null hypothesis – is high). Further, the absence of replication of treatments within blocks precluded estimation of a block × treatment interaction. Such trade-offs are inevitable in hectare-scale, decades-long experiments, however.

All data presented in this paper are publicly available through the Harvard Forest Data Archive (http://harvardforest.fas.harvard.edu/data-archive), in a suite of datasets: HF106 (understory vegetation), HF126 (overstory vegetation), HF161 (litterfall), HF125 (coarse woody debris), HF119 and HF130 (soil respiration), and HF179 (nitrogen pools and dynamics).

Results

Changes in vegetation structure

Overstory trees

Following treatments, the girdled and logged treatments lost 67%–72% of overstory basal area and 61%–71% of overstory stem density (Table 1). Only T. canadensis was affected in the girdled treatment, but girdled individuals ranged from seedlings to canopy trees and they died within 2 years (data in Fig. 4 of Ellison et al., 2010). In contrast, basal area was immediately lost in the logged treatment and included large-diameter T. canadensis, some large Q. rubra and P. strobus, and many smaller A. rubrum and B. lenta (Table 1). By 2009, four years after manipulations, growth of trees in the hemlock and hardwood control treatments resulted in per-plot gains in basal area of 4%–6%; concomitant background mortality led to a per-plot loss of 6%–12% of stems.

Understory vegetation

Understory species richness remained relatively constant in both hemlock control and hardwood treatments over the course of the study, with hardwood treatment plots having the highest herb and shrub richness (Fig. 2a, Table 2). Girdled treatment plots had <10 understory species prior to treatment. Species richness in this treatment increased gradually, resulting in a doubling by 2009 (Fig. 2a, Table 2). Two nonnative species were first identified in the girdled treatment at low abundance by 2007: Berberis thunbergii DC. (Japanese barberry) in the valley girdled plot and Celastrus orbiculatus Thunb. (oriental bittersweet) in the ridge girdled plot. The plots in the logged treatment similarly began with low species richness. In contrast to the girdled treatment, understory species richness increased following logging, but then remained approximately constant for the remainder of the study period (Fig. 2a, Table 2). No nonnative species had recruited into the logged treatment plots by 2009.

Figure 2 Temporal changes in vegetation structure following hemlock removal.

Temporal trajectories of vegetation structural characteristics in the Harvard Forest Hemlock Removal Experiment. Values shown are plot means and standard deviations (where multiple samples were taken in each plot), back-transformed as necessary. Solid lines and symbols are plots in the valley; dashed lines and open symbols are plots on the ridge. Colors indicate treatments: blue – hemlock controls; yellow – all hemlocks girdled; red – hemlocks logged; lavender – hardwood controls.

Table 2 Summary of ANCOVA analyses on vegetation structural characteristics shown in Fig. 2. The models fit were all of the form response variable = β0 + β1 × block + β2 × time + β3 × treatment + β4 × time × treatment; if the response variable was ln-transformed prior to analysis, that is noted in the column heading. Values shown in the first four rows are F-statistics, associated degrees of freedom, and P-values; parameter estimates (SE) for the four treatments – C (hemlock control); G (girdled), L (logged), and H (hardwood control) – are given in the next three rows. Parameter estimates are not back-transformed (for models fit to ln-transformed data). Parameter estimates that are significantly different from 0 are shown in italics.

	Understory richness	Understory cover	ln(Tree seedling density)	ln(Tree seedling cover)	ln(Sapling density +1)	ln(Snag and stump volume)	CWD volume	
Sources of variation								
Intercept	F1,39 = 671.34	F1,47 = 132.35	F1,47 = 2967.94	F1,47 = 0.03	F1,15 = 148.05	F1,15 = 1135.32	F1,15 = 151.22	
P < 0.0001	P < 0.0001	P < 0.0001	P = 0.8615	P < 0.0001	P < 0.0001	P < 0.0001	
Time	F1,39 = 23.65	F1,47 = 21.82	F1,47 = 11.80	F1,47 = 82.57	F1,15 = 1.27	F1,15 = 10.85	F1,15 = 0.00	
P < 0.0001	P < 0.0001	P = 0.0012	P < 0.0001	P = 0.2779	P = 0.0049	P = 0.9867	
Treatment	F3,39 = 53.13	F3,47 = 80.34	F3,47 = 2.55	F3,47 = 32.84	F3,15 = 2.85	F3,15 = 14.53	F3,15 = 43.51	
P < 0.0001	P < 0.0001	P = 0.0668	P < 0.0001	P = 0.0728	P = 0.0001	P < 0.0001	
Time × Treatment	F3,39 = 4.29	F3,47 = 12.58	F3,47 = 4.16	F3,47 = 15.14	F3,15 = 0.49	F3,15 = 7.39	F3,15 = 0.90	
P = 0.0104	P < 0.0001	P = 0.0108	P < 0.0001	P = 0.6929	P = 0.0029	P = 0.4634	
Parameter estimates								
Intercept (β0)	8.38 (1.57)	0.78 (1.62)	10.04 (0.34)	−1.98 (0.33)	3.42 (1.18)	3.22 (0.39)	25.42 (8.24)	
Time (β2)	0.06 (0.58)	0.15 (0.62)	0.18(0.12)	0.22 (0.12)	0.02 (0.39)	0.08 (0.13)	2.89 (2.56)	
Treatment (β3)	G: 0.44 (2.22)	G: 0.49 (2.19)	G: 0.17 (0.42)	G: 0.15 (0.42)	G: 0.55 (1.67)	G: −0.29 (0.55)	G: −2.08 (10.81)	
L: −2.12 (2.22)	L: 0.32 (2.19)	L: 0.21 (0.42)	L: 0.88 (0.42)		L: 0.05 (0.55)	L: 64.45 (10.81)	
H: 17.34 (2.22)	H: 23.71 (2.19)	H: 1.07 (0.42)	H: 3.50 (0.42)	L: 1.71 (1.67)	H: −0.37 (0.55)	H: 18.51 (10.81)	
				H: 1.94 (1.67)			
Time×Treat(β4)	G: 2.49 (0.82)	G: 0.38 (0.88)	G: 0.36 (0.17)	G: 0.73 (0.17)	G: 0.08 (0.56)	G: 0.66 (0.18)	G: −1.82 (3.62)	
L: 2.25 (0.82)	L: 4.59 (0.88)	L: −0.05 (0.17)	L: 0.7 (0.17)		L: −0.10 (0.18)	L: −4.28 (3.62)	
H: 0.68 (0.82)	H: 0.23 (0.88)	H: −0.22(0.17)	H: −0.14 (0.17)	L: 0.61 (0.56)	H: 0.04 (0.18)	H: −5.40 (3.62)	
				H: 0.11 (0.56)			

Understory vegetation cover remained between 1 and 2% in the hemlock control and from 16 to 32% in the hardwood control treatment throughout the sampling period (Fig. 2b, Table 2). Percent cover of understory vegetation increased slowly in the girdled treatment and exceeded cover in the hemlock control treatment by 2009. Understory cover increased more rapidly in the logged treatment, especially after 2007 (Fig. 2b), significantly exceeding cover in both the hemlock control and the girdled treatments, and equaling levels seen in the hardwood control by 2009 (Fig. 2b). The main species driving the increase in understory cover were early successional opportunists and species with long-lived seed banks, including Aralia hispida Vent. (bristly sarsaparilla), Erichtites hieracifolia (L.) Raf. ex DC. (pilewort), Rubus spp. (raspberries and blackberries), and to a lesser extent, Lysimachia quadrifolia L. (whorled loosestrife) and Dennstaedtia punctilobula Michx. (T. Moore) (hay-scented fern).

Tree regeneration

Tree seedling density was low in the hemlock control and logged treatments both before and after canopy manipulations (Fig. 2c); it was nearly 10-fold higher in the hardwood control and this significant difference (Table 2) persisted from 2003–2009. Tree seedlings, especially of B. lenta and A. rubrum increased significantly – to 3.5 × 105 ha−1 – in the girdled treatment through time. Cover of tree seedlings was consistently lowest in hemlock control (<1% cover) and hardwood control (≈5% cover) treatments, but increased slowly and significantly in both girdled (to >40% cover) and logged (to 15% cover) treatments (Fig. 2d; Table 2).

Prior to the manipulations, there were few saplings in any of the plots, and despite some growth, we observed neither significant changes in sapling density through time nor differences in sapling density among treatments (Fig. 2e). The few saplings in the hemlock control treatment were eastern hemlock. Likewise, sapling density was low in the hardwood control treatment throughout the study period; A. rubrum and A. saccharum were the most common sapling species in the valley hardwood plot, whereas A. rubrum and P. strobus were more common in the ridge hardwood plot. The girdling treatment removed all T. canadensis saplings, and even by 2009, most tree regeneration in the girdled plots was still in the seedling (<1.3-m tall) size class and no stems had grown into the sapling size class until 2009. Most saplings in the logged treatment were killed during logging, but stump sprouts of A. rubrum were abundant by 2007 and a few B. lenta had grown from seedlings into saplings on the ridge. By 2009, dense stands (3000–6000 saplings ha−1) of B. lenta saplings covered the logged treatment plots.

Standing and downed dead wood

Volume of stumps and snags was very low in the hemlock and hardwood controls and in the logged treatment plots (Fig. 2f). Volume of stumps and snags in the girdled treatment was similar to both controls in 2003 but then rose significantly (Table 2), by two orders of magnitude, as the girdled trees died (Fig. 2f). Volume of downed CWD in the logged treatment was 2–3 × greater than in any other treatment (Fig. 2g, Table 2). This trend persisted through the five post-treatment years, although CWD volume declined from 2005–2009 as the wood decayed.

Ecosystem function

Litterfall

Litterfall in the hemlock and hardwood controls were not significantly different from one another and remained relatively constant (3– 4 × 103 kg ha−1) over the course of the study (Fig. 3a, Table 3). Total litterfall and hemlock litterfall amounts were significantly affected by hemlock removal (Table 3), and the patterns of change in canopy structure were reflected immediately in litterfall (Figs. 3a, 3b). A strong pulse of litter occurred in the girdled treatment in Spring–Summer 2006, one year after T. canadensis were girdled (Fig. 3a). Hemlock litter comprised >80% of the total litterfall collected in the girdled treatment during April–September 2006 (Fig. 3b). Subsequently, total litterfall in this treatment increased to about the same amount as in the hemlock and hardwood controls by 2009, but was composed mainly of Betula, Quercus and Pinus litter (data not shown). Litterfall in the logged plots was significantly reduced by logging, and slowly increased during the four years after logging to nearly 50% of that observed in the controls (Fig. 3a).

Figure 3 Temporal changes in ecosystem dynamics following hemlock removal.

Temporal trajectories of ecosystem functional characteristics in the Harvard Forest Hemlock Removal Experiment. Values shown are plot means and standard deviations (where multiple samples were taken in each plot), back-transformed as necessary. Solid lines and symbols are plots in the valley; dashed lines and open symbols are plots on the ridge. Colors indicate treatments: blue – hemlock controls; yellow – all hemlocks girdled; red – hemlocks logged; lavender – hardwood controls.

Table 3 Summary of ANCOVA analyses on ecosystem functional characteristics shown in Fig. 3. The models fit were all of the form response variable = β0 + β1 × block + β2 × time + β3 × treatment + β4 × time × treatment; if the response variable was ln-transformed prior to analysis, that is noted in the column heading. Values shown in the first four rows are F-statistics, associated degrees of freedom, and P-values; parameter estimates (SE) for the four treatments – C (hemlock control); G (girdled), L (logged), and H (hardwood control) – are given in the next three rows. Parameter estimates are not back-transformed (for models fit to ln-transformed data). Parameter estimates that are significantly different from 0 are shown in italics.

	ln(litterfall)	ln(hemlock litterfall)	Soil C flux	ln(NH4)	NO3	ln(N mineralization)	
Sources of variation							
Intercept	F1,23 = 10821.55	F1,23 = 3.96	F1,29 = 126.44	F1,47 = 2347.54	F1,47 = 8.35	F1,47 = 397.38	
P < 0.0001	P = 0.0587	P < 0.0001	P < 0.0001	P = 0.0058	P < 0.0001	
Time	F1,23 = 0.14	F1,23 = 6.46	F1,29 = 0.84	F1,47 = 2.53	F1,47 < 0.01	F1,47 = 5.66	
P = 0.7105	P = 0.0182	P = 0.3665	P = 0.1183	P = 0.9922	P = 0.0214	
Treatment	F3,23 = 15.95	F3,23 = 12.45	F2,29 = 5.69	F3,47 = 9.00	F3,47 = 2.44	F3,47 = 2.26	
P < 0.0001	P < 0.0001	P = 0.0083	P < 0.0001	P = 0.0757	P = 0.0933	
Time × Treatment	F3,23 = 3.26	F3,23 = 5.53	F2,29 = 2.05	F3,47 = 2.14	F3,47 = 0.91	F3,47 = 2.39	
P = 0.0399	P = 0.0052	P = 0.1475	P = 0.1075	P = 0.4435	P = 0.0806	
Parameter estimates							
Intercept (β0)	3.86 (0.20)	2.87 (1.16)	0.29 (0.03)	4.67 (0.25)	62.80 (37.73)	−1.04 (0.14)	
Time (β2)	0.00 (0.06)	0.00 (0.33)	0.01 (0.01)	−0.20 (0.10)	−9.77 (11.64)	−0.14 (0.06)	
Treatment (β3)	G: 0.36 (0.27)	G: 2.43 (1.50)	G: −0.02 (0.03)	G: 0.46 (0.35)	G: 15.95 (41.15)	G: −0.05 (0.20)	
	L: −0.94 (0.27)	L: −2.53 (1.50)	L: 0.0 (0.3)	L: 0.84 (0.35)	L: 59.44 (41.15)	L: 0.23 (0.20)	
	H: 0.00 (0.27)	H: −2.37 (1.50)		H: −0.16 (0.35)	H: −17.85 (41.15)	H: −0.09 (0.20)	
Time × Treat (β4)	G: −0.16 (0.09)	G: −1.58 (0.47)	G: 0.02 (0.01)	G: 0.33 (0.14)	G: 25.44 (16.46)	G: 0.20 (0.08)	
	L: 0.12 (0.09)	L: 0.10 (0.47)	L: 0.01 (0.01)	L: 0.11 (0.14)	L: 4.49 (16.46)	L: 0.05 (0.08)	
	H: 0.01 (0.09)	H: −0.22 (0.47)		H: 0.04 (0.14)	H: 8.92 (16.46)	H: 0.05 (0.08)	

Soil respiration

Average seasonal soil respiration dynamics showed some changes as a function of hemlock canopy removal (Fig. 3c), but within-plot variation exceeded among-treatment variation in soil respiration (Fig. 3c; Table 3). By differencing, hemlock roots accounted for approximately 35% of the total soil respiration in intact hemlock stands.

Nitrogen dynamics

Hemlock removal led to transient increases in ammonium (NH4+) and nitrate (NO3−) availability in soils (peaks in Figs. 3d, 3e). As with measures of soil respiration, within-treatment heterogeneity exceeded among-treatment variation in NO3 availability, and neither treatment significantly altered soil NO3 availability (Table 3). Nitrate mobility remained low following cutting or girdling, except for a 2-year pulse beginning in 2008 in the girdled plot on the ridge and beginning in 2007 in the logged plot on the ridge (Fig. 3e).

Across all treatments, net nitrogen mineralization declined significantly through time (Table 3), but within-treatment variation exceeded among-treatment variation throughout the study (Fig. 3f). We observed a small (≈5%), sustained increase in net nitrogen mineralization in the girdled treatment from 2007 to 2009, and a similarly small, albeit transient, increase in net nitrogen mineralization immediately following logging (Fig. 3f).

Discussion

Losses of individual species can have cascading effects on system-wide biological diversity and ecosystem function, but whether specific species have different effects on ecosystem structure function remains an open question that has been addressed much more in theory than in practice (Bunker et al., 2005; Suding et al., 2008; Wardle et al., 2011). It is important to distinguish between effects of loss of dominant (in terms of basal area or biomass) or abundant species and effects of loss of foundation species. For example, American beech is declining rapidly due to beech-bark disease (Houston, 1975; Lovett et al., 2006), but because beech resprouts readily, one consequence of beech-bark disease has been to change the size structure of these forests. Large beech trees are now uncommon, but the number of saplings (sprouts) and even the amount of beech’s basal area in a stand may be much greater than before the occurrence and spread of the disease (Houston, 1975). Similarly, American elm, once a co-dominant in many eastern North American forests, continued to recruit from small trees, which can reproduce before they are killed by Dutch elm disease (Barnes, 1976). Understory composition changes rarely in beech stands infested by beech-bark disease (Twery & Patterson, 1984), and several authors have failed to find expected changes in invertebrate or mammal abundance associated with widespread decline in beech nuts formerly produced by large trees (Faison & Houston, 2004; Garneau et al., 2012). Effects of beech decline on energy and nutrient cycling varies with co-occurring hardwoods, rates of resprouting, and intensity of infestation (Lovett et al., 2006).

In contrast, eastern hemlock has distinctive assemblages of understory plants and animals (Snyder et al., 2002; Tingley et al., 2002; Ellison et al., 2005b; Dilling et al., 2007; Mathewson, 2009; Rohr, Mahan & Kim, 2009; Mallis & Rieske, 2011; Sackett et al., 2011), and affects carbon cycling and hydrological processes differently from both co-occurring hardwoods and co-occurring conifers (Ford & Vose, 2007; Hadley et al., 2008; Brantley, Ford & Vose, 2013). Hemlock, unlike hardwoods, does not resprout, and the hemlock woolly adelgid feeds on all ages and size-classes of hemlock. Thus, there is neither opportunity for rapid regeneration through resprouting nor is there an opportunity for hemlock seedlings to reach maturity and fruit before they are killed by the adelgid. Eastern hemlock, therefore, is a better candidate for a foundation tree species than many other forest dominants. Its decline and death have been hypothesized to lead to both short- and long-term changes in ecological dynamics and ecosystem processes (Ellison et al., 2005a; Lovett et al., 2006).

The Harvard Forest Hemlock Removal Experiment (HF-HeRE) examines this hypothesis by quantifying these changes and testing explicit predictions about how the magnitude and rate of these changes are functions of the mechanism by which a foundation species is lost. In general terms, we predicted that rates of change in biological diversity and ecosystem function would parallel the rate of foundation species loss: slowly when hemlock was girdled (to mimic decline due to infestation by the hemlock woolly adelgid; Yorks, Leopold & Raynal, 2003) but more rapidly when hemlock was cut and removed (to simulate a commercial logging operation; Brooks, 2001). We hypothesize that despite differences in initial rates, changes in vegetation structure and ecosystem function caused by different mechanisms of hemlock loss will converge and come to resemble those seen in the young hardwood stands that represent a plausible scenario of our forests in the future, after hemlock has disappeared from the landscape (Orwig & Foster, 1998). Our results provide strong support for this hypotheses with respect to most measures of vegetation structure, but for fewer measures of ecosystem function.

Changes in vegetation structure

Decline and loss of T. canadensis in the logged and girdled plots at HF-HeRE led to changes in overstory densities and basal area (Table 1) that were similar to those seen in sites long infested by the adelgid (Orwig & Foster, 1998; Orwig, Foster & Mausel, 2002) or that have been salvage logged (Brooks, 2001; Kizlinski et al., 2002). Light availability near ground-level increased gradually over time following girdling but abruptly after logging, followed by a decline with regrowth in the logged treatment (Lustenhouwer, Nicoll & Ellison, 2012). Average daily soil and air temperatures in logged and girdled plots were 2–4 °C warmer in summer and cooler in winter relative to the hemlock or hardwood controls, and both diurnal and seasonal variances in temperatures were highest in the logged treatment (Lustenhouwer, Nicoll & Ellison, 2012). Such changes in light and temperature can strongly impact both vegetation community composition (D’Amato, Orwig & Foster, 2009; Farnsworth, Barker Plotkin & Ellison, 2012) and associated ecosystem properties including decomposition (Berg & McClaugherty, 2008), soil respiration (Savage & Davidson, 2001), and nutrient cycling (Kizlinski et al., 2002).

Removal of T. canadensis by girdling or logging resulted in a 2- to 3-fold increase in species richness after either treatment. Consistent with our first prediction, understory cover in the girdled treatment plots increased slowly (Figs. 2a, 2b) because overstory trees died slowly and the majority of snags were still standing and providing partial shade 4–5 years after the canopy manipulation treatment had been applied. We anticipate that understory vegetation in this treatment will continue to increase in cover and species richness. In contrast, understory vegetation cover in the logged treatment plots increased rapidly and matched total cover in the hardwood control plots by 2009 (Figs. 2a, 2b). Shade intolerant species including Rubus spp., Aralia hispida, and Carex spp. initially were absent in all six T. canadensis-dominated plots, but established from both the seed bank and the seed rain in soils scarified by logging (Farnsworth, Barker Plotkin & Ellison, 2012) and grew quickly in these scarified areas. Similar increases in total richness and cover have been observed following girdling (Yorks, Leopold & Raynal, 2003; Ford et al., 2012) or salvage logging (Kizlinski et al., 2002; D Orwig, unpublished data) of T. canadensis elsewhere. However, the heavy recruitment of birch (Betula spp.) into the sapling layer within four years of girdling has not been observed in other girdling studies (Yorks, Leopold & Raynal, 2003; Ford et al., 2012), perhaps due to lower deer browsing and lack of rhododendron (Rhododendron maximum L.) cover that limit rapid recruitment south of our study areas. These results overall highlight the fact that healthy hemlock act as an ecological filter, limiting seedling and understory plant establishment (Rogers, 1980; D’Amato, Orwig & Foster, 2009; Orwig et al., 2012). Now that the adelgid has colonized the hemlock control plots, however, they are also poised for change, and will provide important comparisons with responses observed following girdling.

Changes in standing and downed dead wood

Changes in coarse woody debris volume (Figs. 2f, 2g) were consistent with our second prediction. By the end of 2009, most dead trees were still standing in the girdled treatment plots. Once they fall, however, volume of fallen wood will more than double the levels currently found in the logged plots. Ironically, although the hemlock canopy is lost, this large input of CWD onto the soil surface will bring the dead wood structure of this treatment closer to that seen in old-growth T. canadensis stand structure (D’Amato, Orwig & Foster, 2008) than to the volume of standing dead wood or CWD in young hardwood stands. These fallen boles likely will provide safe sites for seedling establishment and cover for amphibians (Mathewson, 2009) and, as they decompose, also will slowly release nutrients into the soil.

Changes in ecosystem functions

Our third prediction was supported most clearly for changes in litterfall, an index of aboveground primary productivity (Zheng, Prince & Hame, 2004). In the girdled treatment plots, there was a sharp pulse in litterfall followed by a gradual decline (Figs. 3a, 3b; see also Yorks, Leopold & Raynal, 2003; Nuckolls et al., 2009). This is a transient loss of standing biomass, which then recovered to pretreatment levels as vegetation colonized or regrew in the experimental plots. Total litterfall following girdling recovered within four years to levels observed in both hemlock and hardwood controls as Pinus strobus, Quercus spp. and Betula spp. growth increased. Very similar patterns and total amounts of litterfall were observed four years after girdling hemlock in southern Appalachian forests (Knoepp et al., 2011). In the logged treatment plots, litterfall immediately decreased following logging then slowly increased. By 2009, however, the amount of litterfall in the logged plots treatment was still lower than in the girdled treatment or either of the two controls (Figs. 3a, 3b).

Contrary to our third prediction, variation in soil respiration, nitrogen availability, and nitrogen cycling generally was higher within treatments and years than among treatments or years, and any responses to treatments were modest and transient. Any initial differences among treatments rapidly recovered to pre-treatment levels (Figs. 3c–3f). Our approximate autotrophic respiration rate estimates of 36%–46% are similar to the 48% measured by Gaudinski et al. (2000) at the Harvard Forest using isotopic analysis of respired 14C. We measured 43% autotrophic respiration using the same isotopic analysis methodology as Gaudinski et al. (2000) within the Simes girdled treatment plots (K Savage & E Davidson, unpublished data). Similar ranges attributed to autotrophs have been estimated in other studies (Hanson et al., 2000; Levy-Varon, Schuster & Griffin, 2012).

Despite the dramatic changes caused by girdling and logging in microenvironmental conditions (Lustenhouwer, Nicoll & Ellison, 2012), vegetation structure (Fig. 2; Table 2), and productivity (Figs. 3a, 3b; Table 3) and the transient shifts in carbon dynamics (Fig. 3c; Table 3), our experimental treatments resulted in only modest, short-term changes in nitrogen cycling (Figs. 3d–3f; Table 3). Ammonium availability in the girdled treatment plots did not increase until two years after girdling and peaked one year later, a result expected because T. canadensis trees did not die or drop their needles immediately. The short-lived duration of nutrient capture on resins is likely related to the rapid regrowth of vegetation in the logging treatment (see also Templer & McCann, 2010). Short-lived increases in ammonium and nitrate availability also have been observed in other logging and girdling studies (Kizlinski et al., 2002; Yorks, Leopold & Raynal, 2003; Nave et al., 2011), and in adelgid-infested forests (Jenkins, Aber & Canham, 1999; Orwig et al., 2008). Net nitrogen mineralization was not significantly affected by logging or girdling, a result also seen other hemlock studies in girdled (Knoepp et al., 2011) and logged stands (Kizlinski et al., 2002), and consistent with findings following a substantial physical disturbance (simulated hurricane; Bowden et al., 1993). However, we also found no substantive differences in nitrogen mineralization between hemlock and hardwood control plots. The lack of major differences in soil pH or forest floor C:N is consistent with these findings, which are also supported by recent meta-analyses (Mueller et al., 2012).

We caution, however, that it may take much longer than a decade or two for changes in soil dynamics resulting from the loss of hemlock to be manifest (DJ Lodge, personal communication, 24 September 2004). One of the dominant drivers of soil dynamics – decomposition of large fallen boles and other coarse woody debris – is very different in hemlock (and other conifer-dominated) stands than in hardwood dominated stands. In the former, brown-rot fungi dominate, and they primarily decompose cellulose. In the latter, white-rot fungi dominate, and they primarily decompose lignin; in general, white-rot fungi are much more efficient (and rapid) decomposers (Hatakka, 2001; Floudas et al., 2012). We predict that soil nutrient availability will decline significantly only once dead hemlock boles and smaller coarse hemlock woody debris have decomposed and brown-rot fungi disappear, but this may take one or two centuries.

Conclusions

Loss of the foundation tree species, T. canadensis, by either girdling or logging, leads to short- and long-term changes in vegetation structure and ecosystem function. Rapid removal by logging leads to abrupt, rapid changes, whereas girdling (and by inference, the adelgid itself) causes slower but no less important responses of similar magnitude several years later. Vegetation richness, cover, and density increase continuously following hemlock removal and exert strong, potentially stabilizing, biotic control on the fluxes of nutrients. Thus, these ecosystem processes exhibited short-term fluctuations following T. canadensis removal but recovered to near pre-treatment levels within four years, highlighting the resilience – at least in the short-term – of some forest ecosystem processes to disturbances (Bowden et al., 1993; Foster et al., 1997). Results from HF-HeRE, together with results from observations and experiments on other foundation species suggest that their continued losses, together with human responses to ongoing environmental changes, may have profound impacts on the structure and function of forested ecosystems for decades to come.

Mackenzie Bennett, Peter Bettman-Kerson, Charlie Boyd, Naomi Clark, David Franklin, Clarisse Hart, Sultana Jefts, Susan Irizarry, Alanna Kassarjian, Bennet Leon, Natalie Levy, Samantha Marshall, Nicole Mercier, Jan Ng, Amanda Parks, Samantha Petit, Haley Smith, Relena Ribbons, Christy Rollinson, Ernesto Rodríguez, Tawny Virgilio, Kelly Walton, Matt Waterhouse, and Kristin Williams assisted in the field work. Elizabeth Farnsworth, David Foster, two anonymous reviewers, and academic editor Michael Huston provided helpful comments on early versions of the manuscript.

Additional Information and Declarations

Competing Interests

Author Contributions

Field Study Permissions

Data Deposition

AM Ellison is an Academic Editor for PeerJ.

David A. Orwig, Audrey A. Barker Plotkin and Kathleen E. Savage conceived and designed the experiments, performed the experiments, analyzed the data, contributed reagents/materials/analysis tools, wrote the paper.

Eric A. Davidson conceived and designed the experiments, analyzed the data, wrote the paper.

Heidi Lux performed the experiments, contributed reagents/materials/analysis tools, wrote the paper.

Aaron M. Ellison conceived and designed the experiments, performed the experiments, analyzed the data, wrote the paper.

The following information was supplied relating to ethical approvals (i.e. approving body and any reference numbers):

Permit to work in wetlands and/or bordering vegetated wetlands (applies only to Plot 1 of the experiment) issued by the Petersham Conservation Commission. In a letter dated 14 June 2005, the Chairman of the Petersham Conservation Commission, Robert A. Clark, wrote: “the wetlands will not be disturbed other than the dynamics of changes in tree species and that the study will not negatively alter the wetlands. The Commission agreed that the filing of a Notice of Intent is not required.”

The following information was supplied regarding the deposition of related data:

All data presented in this paper are publicly available through the Harvard Forest Data Archive (http://harvardforest.fas.harvard.edu/data-archive), in a suite of datasets: HF106 (understory vegetation), HF126 (overstory vegetation), HF161 (litterfall), HF125 (coarse woody debris), HF119 and HF130 (soil respiration), and HF179 (nitrogen pools and dynamics).

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
