# Peer review of "Foundation species loss affects vegetation structure more than ecosystem function in a northeastern USA forest"

_PeerJ, doi:10.7717/peerj.41_

## Round 0.1 · original submission · Minor Revisions

While one of the reviewers was not very enthused about the novelty of your results, it is quality, not novelty, that is the main criterion for PeerJ. The other reviewer makes a number of very good suggestions about how the manuscript can be improved. In particular, I expect you to include more background and comparison with the impacts of other major tree diseases that have affected the forests of the Northeast. I also believe you can tighten up the prose and remove some unnecessary description and duplication as suggested by the reviewer.

Reviewer 1 ·

Basic reporting

Only one comment: Since rationale for paper rests on understanding consequences of loss, specifically, of foundation species, it seems to me that there should be some recognition of the relatively large literature on consequences of loss of American beech due to disease. There are published studies that address at least some of the questions posed here about community/ecosystem effects, and this is in the same/similar ecosystem. There's also some work on ecosystem/structural consequences of loss of American elm in wetland forests in same region.

Experimental design

Generally sound in design, but a few things to note (these are detailed in notes for authors)

Biggest conceptual/structural issue: 'Hypotheses' given as motivating the study are not mechanistic hypotheses, but predictions that are given without explaining the mechanistic reasoning behind them. It would be better to present hypotheses as functional relationships/mechanisms suspected to be at work, and then to derive these predictions from them as means of testing hypotheses....

Methods:
- there are issues around interpreting effects of logging as due to hemlock removal when other trees were removed as well (see comment line 289 and elsewhere).
- Need to clarify issue in comment on L 251 re estimation of aboveground NPP; the conclusion as stated in lines 520-521 does not follow for me; the 'sharp pulse' can't be read as reflecting an increase in aboveground PP as implied by this language; it's break-down of accumulated pool of canopy structural biomass...

Validity of the findings

My only issues here concern presentation: Results section are unnecessarily 'prosey' repeating details in text that are obvious in tables/figures and not important in emphasizing main points, telling story. Discussion tends to repeat details from Results unnecessarily. But also does good job of synthesis and putting things in context of literature...

Additional comments

These line-by-line comments complement and elaborate on the general points above. I also note that I think the whole paper tends to be a bit more discursive than need be, and migth be shortened to make for a more compact 'read' without loss of substance.

l23: "remains an open question whether particular species with particular characteristics will disproportionately change how ecosystems function" It remains an important question, but it would be appropriate to recognize that community/system consequences of loss of specific species HAVE been addressed for some cases -- maybe most relevant here, a number of studies on consequences of loss of beech due to beech-bark disease -- another late-successional 'foundation species' in similar forests... Quite a few papers over last 30 years...
(Odd that beech not mentioned in next paragraph, since it's closer parallel in many ways than the cases mentioned)

l38: extra "and"

l98: 'a' rather than 'an'

l96-103: This sentence needs reworking -- probably breaking up into two or more sentences, maybe a little expansion (especially as it lays out basic conceptual structure). Currently diagrams as 'experiment compares and contrasts rates.... to two mechanisms...', which isn't quite right.

l105: 'following' rather than 'caused by'? attribution of causality needs to follow design that justifies it...

l107: "the hypotheses": These are predictions, but don't really spell out reasoning. I'd prefer to see 'hypotheses' stated in terms of processes or causal relationships being focused on, and these kinds of pattern predictions stated as testable predictions of those hypotheses... WHY are these expectations reasonable?

L118: this prediction is more far-reaching and less obviously reasonable than first three. Convergence with hardwood controls seems to requrie several assumptions that aren't really addressed here.

L177-178: The rationale for cutting of hardwoods and white pine is not clear, since experiment is directed at detecting effects of removal of hemlock? -- ah, I see from subsequent paragraph that it's intended to mimic a salvage operation. Okay, but that means predictions given in intro have to be in terms of more complex differences between treatments -- not just whether hemlock is felled or dies standing, but whether other stuff is killed/removed or not...

L210: Grasses and sedges to genus only: This is too bad, since sedges can constitute a good chunk of the herbaceous diversity in such forests...

l251: litterfall as index of ANPP: This seems okay EXCEPT for one important issue; does it include disintegrating twig/branch material from girdled trees on the girdled plots? That would NOT be appropriate as a component of current ANPP -- would lead to overestimation of post-treatment ANPP for a few years as canopy structure breaks up!

l288-90: Back to question about cutting pine and hardwoods; in the logged plot, doesn't difference represent MORE than contribution of hemlock (i.e., that of other trees cut)?

l300-301: Yes, but that heterotrophic respiration is likely to be enhanced by breakdown of roots of killed trees in experimental plots?

l343-344: 'increased abruptly.. remained stable' This is not visually evident for both logged plots; one instance COULD be interpreted this way; the other looks pretty gradual to me with increase continuing in the last sample interval

l410-424: This is an example of over-description; I'd suggest reducing amount of verbal retelling of what's clear from tables/figures -- especially when there's no meaningful pattern relative to hypotheses/predictions...

l448-449: It might be possible to draw some something from literature on response to beech bark disease mortality on this front...

l522; this 'pulse' should be interpreted as related to increased ANPP as implied by the foregoing sentence; it's a transient loss fo a biomass reservoir and that should be clearly distinguished...

FIGURE 2: There needs to be an in-graph legend matching treatment to color.

Reviewer 2 ·

Basic reporting

So, let me think about this. If I logged a stand and removed big trees, what would I expect? Let's see - decreased basal area, decreased cover, more CWD on the ground from felling the trees, less litterfall (because there are fewer big trees), lower soil C and N flux because of decreased litter input but slowly recovering as the canopy closed and litterfall recovered. And if I girdled trees but left them standing? Smaller changes in the above properties.

This is what this experiment demonstrated.

So what new things have we learned?

Experimental design

The experimental design was fine. I just don't think that the results teach us anything new.

Validity of the findings

See comments in other boxes.

Additional comments

This experiment was obviously a lot of work and well-executed, but I don't find any of the results to be all that surprising. If you wanted to examine specifically what the loss of hemlock did compared to the loss of other "foundation species", the experiment should have been repeated removing another species with very different potential effects on the ecosystem, such as sugar maple. But that would have been a massive experiment.

---

## Round 0.2 · accepted · Accept

I appreciate your careful attention to the suggestions from the reviewers, and am grateful that two of the reviewers provided such detailed and constructive comments. The paper is much improved.

However, as I read through your "track-changes" copy I noticed a number of careless typographical errors. There are probably others that I did not notice.

Please carefully proof-read the final version that you submit for publication, correcting all the errors I noticed, as well as any similar problems that I didn't notice (I was not proof-reading, just scanning).

The following line numbers are from the track-changes copy, but the one that I checked (line 217) was also in the pdf (line 228). Presumably you can locate the others.

217 flowers NOT flowerd (damn querty)
490 delete "at least"
507 word missing - decline and ????
543 word missing 4-5 [years?]

---

## Author Rebuttal · Round 0.2

**HARVARD UNIVERSITY**

**HARVARD FOREST**

**324 NORTH MAIN STREET**
**PETERSHAM, MASSACHUSETTS**
**U.S.A.   01366**

[Figure]

**PHONE: 978-724-3302**
**FAX: 978-724-3595**

21 January 2013

Dr. Michael Huston, Academic Editor
Dear Michael,

Thank you for getting two constructive reviews of our manuscript "Foundation species loss affects vegetation structure more than ecosystem function in a northeastern USA forest" (#2012:11:32:0:1:REVIEW). We appreciate the care with which the reviewers assessed the manuscript, and we have revised it following their, and your, suggestions.

Per your instructions, we are submitting this rebuttal letter, a .docx version of the manuscript with tracked changes, and a clean, un-tracked version for production purposes.

In the list of changes detailed on the following pages, please note that the line numbers refer to the numbers in the tracked-changes version of the .docx version of the manuscript submitted through the *PeerJ* system. I note that the pdf conversion of the original submission both changed the line-numbering relative to the .doc file submitted *and* dropped line numbers as noted by Reviewer 3.

Thank you for considering our work for publication in *PeerJ*. This is an exciting approach to open-access publication, and we are delighted to be participating in it.

Your sincerely,

//Aaron//

Aaron M. Ellison
*Senior Research Fellow*

**Comments from the Academic Editor**

While one of the reviewers was not very enthused about the novelty of your results, it is quality, not novelty, that is the main criterion for PeerJ. The other reviewer makes a number of very good suggestions about how the manuscript can be improved. In particular, I expect you to include more background and comparison with the impacts of other major tree diseases that have affected the forests of the Northeast. I also believe you can tighten up the prose and remove some unnecessary description and duplication as suggested by the reviewer.

The concerns of the third reviewer (attached below) involve clarity of terms and consistency of definitions, as well as technical questions that need to be addressed. Addressing these issues will greatly improve the manuscript's clarity and impact.

We appreciate your attention to the mission and scope of *PeerJ*. We have added additional citations to chestnut blight, Dutch elm disease, and beech-bark disease in the Introduction (lines 38-44 **of the tracked-changes version**), and expanded on this topic in the Discussion (lines 482-509). The latter (Discussion) also emphasizes similarities in responses of these hardwoods to disease and differences between the effects of the loss of these forest dominants and the loss of a hypothesized foundation species – eastern hemlock.

Tightening up of prose and clarification of terms &c. are discussed in more detail in our response to each point of the reviewers.

**Reviewer 1**

**Basic reporting**

Only one comment: Since rationale for paper rests on understanding consequences of loss, specifically, of foundation species, it seems to me that there should be some recognition of the relatively large literature on consequences of loss of American beech due to disease. There are published studies that address at least some of the questions posed here about community/ecosystem effects, and this is in the same/similar ecosystem. There's also some work on ecosystem/structural consequences of loss of American elm in wetland forests in same region.

As discussed above, we have added citations to beech-bark disease and Dutch elm disease in the Introduction, and expanded on these points in the Discussion.

**Experimental design**

Generally sound in design, but a few things to note (these are detailed in notes for authors)

Biggest conceptual/structural issue: 'Hypotheses' given as motivating the study are not mechanistic hypotheses, but predictions that are given without explaining the mechanistic reasoning behind them. It would be better to present hypotheses as functional relationships/mechanisms suspected to be at work, and then to derive these predictions from them as means of testing hypotheses....

We have changed the wording throughout to differentiate between "Hypotheses" (lines 32, 52, 131, 508, 510, 517) and "Predictions" (lines 114, 211, 239, 255, 511, 539, 564, 575, 588). Note especially

Methods:
- there are issues around interpreting effects of logging as due to hemlock removal when other trees were removed as well (see comment line 289 and elsewhere).

We now discuss this explicitly (lines 195-200)

- Need to clarify issue in comment on L 251 re estimation of aboveground NPP; the conclusion as stated in lines 520-521 does not follow for me; the 'sharp pulse' can't be read as reflecting an increase in aboveground PP as implied by this language; it's break-down of accumulated pool of canopy structural biomass...

clarified (lines 578-579)

**Validity of the findings**

My only issues here concern presentation: Results section are unnecessarily 'prosey' repeating details in text that are obvious in tables/figures and not important in emphasizing main points, telling story. Discussion tends to repeat details from Results unnecessarily. But also does good job of synthesis and putting things in context of literature...

See responses below to line-by-line comments

**Comments for the author**

These line-by-line comments complement and elaborate on the general points above. I also note that I think the whole paper tends to be a bit more discursive than need be, and migth be shortened to make for a more compact 'read' without loss of substance.

l23: "remains an open question whether particular species with particular characteristics will disproportionately change how ecosystems function" It remains an important question, but it would be appropriate to recognize that community/system consequences of loss of specific species HAVE been addressed for some cases -- maybe most relevant here, a number of studies on consequences of loss of beech due to beech-bark disease -- another late-successional 'foundation species' in similar forests... Quite a few papers over last 30 years...
(Odd that beech not mentioned in next paragraph, since it's closer parallel in many ways than the cases mentioned)

changed. see especially lines 38-44, 492-509

l38: extra "and"

deleted

l98: 'a' rather than 'an'

changed

l96-103: This sentence needs reworking -- probably breaking up into two or more sentences, maybe a little expansion (especially as it lays out basic conceptual structure). Currently diagrams as 'experiment compares and contrasts rates.... to two mechanisms...', which isn't quite right.

reworked. Lines 99-107.

l105: 'following' rather than 'caused by'? attribution of causality needs to follow design that justifies it...

changed.

l107: "the hypotheses": These are predictions, but don't really spell out reasoning. I'd prefer to see 'hypotheses' stated in terms of processes or causal relationships being focused on, and these kinds of pattern predictions stated as testable predictions of those hypotheses... WHY are these expectations reasonable?

changed, as noted above.

L118: this prediction is more far-reaching and less obviously reasonable than first three. Convergence with hardwood controls seems to requrie several assumptions that aren't really addressed here.

We've taken this out of the predictions and reframed it as a hypothesis (lines 131-133).

L177-178: The rationale for cutting of hardwoods and white pine is not clear, since experiment is directed at detecting effects of removal of hemlock? -- ah, I see from subsequent paragraph that it's intended to mimic a salvage operation. Okay, but that means predictions given in intro have to be in terms of more complex differences between treatments -- not just whether hemlock is felled or dies standing, but whether other stuff is killed/removed or not...

We appreciate this astute observation, and note that it's the first time a reviewer of our now nearly a dozen papers on this experiment has commented on it. See additional text in lines 195-200.

L210: Grasses and sedges to genus only: This is too bad, since sedges can constitute a good chunk of the herbaceous diversity in such forests...

Yes it is too bad, but no flowers, no fruits, and the taxonomy is a mess.

l251: litterfall as index of ANPP: This seems okay EXCEPT for one important issue; does it include disintegrating twig/branch material from girdled trees on the girdled plots? That would NOT be appropriate as a component of current ANPP -- would lead to overestimation of post-treatment ANPP for a few years as canopy structure breaks up!

It does not include disintgrating twig/branch material. It is virtually all hemlock needles and/or cones

l288-90: Back to question about cutting pine and hardwoods; in the logged plot, doesn't difference represent MORE than contribution of hemlock (i.e., that of other trees cut)?

Maybe (see lines 195-200), but it seems redundant to repeat it here.

l300-301: Yes, but that heterotrophic respiration is likely to be enhanced by breakdown of roots of killed trees in experimental plots?

This was already noted (and still is, lines 297-299)

l343-344: 'increased abruptly.. remained stable' This is not visually evident for both logged plots; one instance COULD be interpreted this way; the other looks pretty gradual to me with increase continuing in the last sample interval

Language changed (lines 373, 381, 382)

l410-424: This is an example of over-description; I'd suggest reducing amount of verbal retelling of what's clear from tables/figures -- especially when there's no meaningful pattern relative to hypotheses/predictions...

Agreed. We have deleted lines 445-454, 460-465, 593-602, and 608-613.

l448-449: It might be possible to draw some something from literature on response to beech bark disease mortality on this front...

probably not, but see lines 492-509.

l522; this 'pulse' should be interpreted as related to increased ANPP as implied by the foregoing sentence; it's a transient loss fo a biomass reservoir and that should be clearly distinguished...

agreed; see lines 578-579.

FIGURE 2: There needs to be an in-graph legend matching treatment to color.

added.

## Reviewer 3

General comments

The manuscript is based on a significant amount of work coming from an experimental manipulation. It is relatively early in the experiment in terms of ecological response, but I believe the results are worth publishing at this point. The manuscript is generally well written, but there are some problems that need to be addressed.

There is an inherent circularity in how the way foundational species is being used by the authors. This is a bit hard to avoid, but as the results of the study indicate foundational species may not be so foundational for some ecological processes and functions. So we read that hemlock is a foundational species in terms of structure, but not for some ecosystem processes. But then again it is repeatedly referred to as a foundational species.

We now present this as a hypothesis, and as a "putative" foundation species (see lines 52, 56, 506-511).

Another problem is that the authors tend to over generalize one set of structures to all structures and a few ecosystem processes to all ecosystem processes. Therefore I feel that the manuscript could be strengthened by qualifying these statements.

Qualified, esp. p. 650-653.

I was a bit surprised to see little in the discussion on statistical power and type II errors. I understand these kinds of experiments are difficult to replicate. However, with just two replicates of each treatment the power of the statistical tests was very low. Therefore type II errors would be a real possibility, particularly for the ecosystem variables examined. I understand that regardless of the statistical power of the tests the changes in ecosystem parameters was small than for the structural ones. That makes sense in that the treatments directly impacted vegetative structure and indirectly impacted ecosystem processes. Further that some of the ecosystem functions might have been more buffered.

Some text added, lines 344-350.

I believe the authors would benefit by including the concept of biological legacies in their thinking. It seems to me that there is greater response to the treatments the lower the biological legacy left.

We're not sure to what the reviewer is referring, so we didn't do anything with this comment.

One issue that concerns me in the conclusions: it is stated the loss of hemlock leads to increases in understory cover (although it is not qualified and could be tree canopy cover) and diversity. But would not understory cover have increased if any of the major tree species had been removed. And does this mean that the removal of white pine and red oak in the logged plots did not cause the increase in cover of understory plants. Would that occurred regardless of species removed? I suppose a better case could be made for diversity of understory plants, but even here would that not have increased regardless of which species was removed? Does that mean all major tree species are foundational species? If the authors could clarify their thinking it would be most helpful.

We've tried, throughout.

I was a bit surprised to see the methods and results described in present tense. I believe these are all supposed to be in past tense given they have either occurred or were observed and are currently not ongoing. Correcting this issue will involve considerable revision as it occurred in many sections of the manuscript.

This was a stylistic attempt to distinguish between measurements that are ongoing (even now) and those that had a fixed endpoint. But it clearly didn't work, so we've changed all the text to past tense.

Specific comments (line)

Abstract

1 If the first sentence is true and hemlock is a foundational species, then one has to wonder why the study was needed in the first place. Clearly the study is checking to what degree hemlock is a foundational species. So some revision indicating that this is an open question would be helpful to the reader and more convincing.

Modified Abstract line 5 (**in tracked version**)

8 the manuscript describes CWD in various ways. In some places as here it is evidently snags and logs, but other places it is logs only. This inconsistency is quite confusing. The most general class is CWD, more specific ones are snags (standing dead), logs (downed dead), and stumps. It would be helpful to use that framework.

In Abstract and throughout, we have followed the guidance of this reviewer.

10 What is included in cover? It is not clear and could be of the understory plants or the canopy or both.

understory vegetation added, line 10.

Main body of text

21 Starting sentences with but should not be done on a routine basis. How about using "however"?

changed

27 this sentence starts general and then jumps to forests. It really is confusing to do that. I suggest the two ideas be placed in separate sentences.

changed

30 Here is an example of circular logic that needs to be avoided if at all possible. If a species is gone then yes, it is gone. If a foundational species is defined as influencing the ecosystem, then yes its loss will influence the ecosystem. Isn't the real question "to what degree is this true?"

modified

92 I have doubts that experiments are the only way. I agree they are a very good way, but not the only way. And doesn't this depend on the experimental design and degree of confounding of effects? Surely some experiments are just as misleading as observational studies.

This is a philosophical discussion that we're not going to delve into here.

109 It needs to be made clear in the abstract what is meant by cover.

done

114 earlier it was implied that CWD was logs, snags, and stumps. Now it is just logs. It would be better to be consistent and not use general terms where a specific one is needed.

changed lines 119-120.

119 while I agree that some long-term convergence is likely to occur, I am wondering how a short-term study could possible test this.

It can't, so this is a hypothesis. lines 131-133.

234 fallen CWD is fine to use, but by shortening this later to CWD is becomes very confusing as standing dead wood is also (at least some of it CWD). Standing dead trees are snags. A clearer, consistent terminology should be used.

changed lines 239-241.

239 here is the first of many instances of using present tense for measurements that occurred in the past.  These all need to be changed.

all changed.

246 there are no methods to calculate mass of CWD, yet there are results on mass.

we meant volume. changed

249 this section is also written in present tense.

all changed.

289 I am not so sure it is as simple as this.   Root exudates would cease when roots die, and that carbon would be respired by microbes and hence be RH.

292 I don't see how one could state that this is a conservative estimate of RA.  For one thing there is the issue of root exudates mentioned above. And as the authors indicate root exudates might lead to priming and that might lead to a reduction in RH as well as RA.   While the difference is an estimate I have no idea if it is liberal or conservative.  Citing references would help to see which is more likely, but it is not nearly as simple as stated.

301 Do the authors mean RH for the entire ecosystem? That could not be close to true unless there is no loss of carbon from the dead wood created by the treatments. Do the authors have evidence that the dead wood created by logging and girdling has not decomposed?

We are trying to be as circumspect as possible with our estimation of heterotrophic soil respiration, and references are all there. lines 291-304.

For some reason the line numbers cease on the manuscript.

software bug at PeerJ

The entire soil section seems to be in present tense.

all changed

331 What is the mortality on an annual basis?  This is a more useful set of numbers although cumulative mortality would be worth reporting.

We only measure it every 5 years, so we don't have it on an annual bases. That's why we report the data we do.

337 it would be helpful to have the numbers of species presented.

There in the figure. As per Reviewer 1, why repeat them in text?

373 study is a bit vague.  Is it the study period or study area? That is time or space? It could be either.

period.

389 is the CWD downed as the subtitle suggests?  Or is this the total of downed and standing dead wood? It is not very clear.  What were the totals for the standing and downed wood? Of course if one cuts down all the dead trees the amount snags is low, but it might not influence the total amount of dead wood.

391How was the mass loss determined? There are no methods on this and what exactly were the values.

language revised, lines 418-425.

418 Is this carbon efflux from the soil or the total ecosystem? These are not the same thing at all.  The authors note that the CWD lost mass, so how would soil respiration reflect the total carbon efflux from the forest. This is pretty sloppy carbon accounting.

Soil respiration.

508 Aren't snags a form of CWD?  The subtitle on line 507 has a clear terminology, but this does not. In fact it is not consistent at all with line 507.

517 Do the authors mean to imply that the standing dead wood or snags and stumps are not decomposing and hence losing carbon?  As written the dead wood needs to fall to the ground before decomposition begins. Is that really true in this forest?  If so it should be documented.

language modified lines 567-572.

529 Can the authors provide any estimates of stem/branch related production to put the litterfall numbers in some context.  It is unlikely that stem/branch related production has recovered anywhere close to the foliage one.

good question, we have no idea.

536 Correct me if I am wrong but hypothesis 4 was really about long-term convergence. Since the convergence happened in the short-term I am not so sure your observations support it.

agreed, changed.

557 perhaps this could just be root systems?  Not sure what rooting adds.

agreed, deleted.

584 Unfortunately the authors have completely mixed up the effects of brown- and white-rots.  White-rots degrade lignin and brown-rots do not.

switched. Doesn't change the conclusion, which was based on the correct use of brown and white rots.

596 Wouldn't vegetation cover and richness have increased if any species had been removed?  The experimental design would have had to have removed hardwoods to have tested this.

and we would have needed a few dozen more hectares. Next experiment...

598 this needs to be qualified to the ecosystem processes that were studied. There are other ecosystem processes that were  not studied and they could have responded differently.

changed.